# Cross-feeding promotes heterogeneity within yeast cell populations

Kevin K. Y. Hu [1], Ankita Suri[1], Geoff Dumsday[2] & Victoria S. Haritos [1] ✉

Cellular heterogeneity in cell populations of isogenic origin is driven by intrinsic factors such as stochastic gene expression, as well as external factors like nutrient availability and interactions with neighbouring cells. Heterogeneity promotes population fitness and thus has important implications in antimicrobial and anticancer treatments, where stress tolerance plays a significant role. Here, we study plasmid retention dynamics within a population of plasmid-complemented *ura3Δ0* yeast cells, and show that the exchange of complementary metabolites between plasmid-carrying prototrophs and plasmid-free auxotrophs allows the latter to survive and proliferate in selective environments. This process also affects plasmid copy number in plasmid-carrying prototrophs, further promoting cellular functional heterogeneity. Finally, we show that targeted genetic engineering can be used to suppress cross-feeding and reduce the frequency of plasmid-free auxotrophs, or to exploit it for intentional population diversification and division of labour in co-culture systems.

Metabolite exchange is a ubiquitous phenomenon, occurring not only among unicellular species but also between cells within multicellular organisms[1,2]. Often, surrounding a microbial community, we can find a nutrient-rich extracellular metabolome that includes glycolytic- and tricarboxylic acid cycle-derived intermediates, amino acids, nucleotides, pro-survival factors like reduced glutathione (GSH) and glycerol, as well as other overflow metabolites such as ethanol and acetate[3–5]. This shared pool of resources provides constituent members with a degree of freedom to undergo metabolic shift and division of labour, resulting in a more diversified and heterogeneous community consisting of both self-sustaining prototrophs and auxotrophs that have lost the essential metabolic pathway(s) required for survival[6–8].

The development of metabolically differentiated subgroups within the population, in turn, provides an effective defence mechanism against unexpected adverse changes in environmental conditions and other external insults; for instance, amino acid auxotrophs are generally more drug resilient compared to prototrophs[9]. In nosocomial settings, metabolite cross-feeding between pathogens was shown to promote biofilm formation[10], enhance the growth and virulence of antibiotic-resistant bacterial strains[11], as well as conferring cross-protection against antimicrobials to co-infecting pathogens[12],

altogether highlighting metabolite exchange as a crucial factor in achieving effective treatment strategies. In another clinical context, exogenous amino acid dependencies have also been observed in certain cancer cell types, which either are related to their cell of origin or as a result of them undergoing profound metabolic reprogramming to accommodate their enhanced proliferation[13]. Such metabolic vulnerabilities have been exploited for the treatment of paediatric acute lymphoblastic leukaemia[14], while multiple novel amino acid depletion cancer therapies are also currently under clinical evaluation (reviewed in[13]).

In bioengineering, auxotrophic bacteria are known to effectively perform intercellular metabolite exchange with co-existing cells to overcome their own metabolic deficiencies, either being unilaterally supported by prototrophs[15,16] or forming syntrophic relationships with other auxotrophs[17–19]. On the other hand, while yeasts are capable of supporting the growth of co-cultivated fastidious bacteria through essential metabolite sharing[20], the success rate of establishing syntrophic growth between complementary yeast auxotrophs is very low[21], unless the cells were genetically engineered with feedback-resistant mutations to overproduce the complementary metabolites[22–24]. Additionally, while auxotrophic prokaryotes are

[1]Department of Chemical and Biological Engineering, Monash University, Clayton, VIC 3800, Australia. [2]Commonwealth Scientific and Industrial Research Organisation, Clayton, VIC 3169, Australia. ✉e-mail: victoria.haritos@monash.edu

prevalent in natural microbial communities[7], natural yeast isolates are generally prototrophic[25,26]. Therefore, one long-standing question concerning metabolite exchange between eukaryotic microbes like yeasts is whether metabolite exchange could be performed to an extent that would compromise the retention rate of plasmid DNA (pDNA) that is maintained through auxotrophic selection within a recombinant population. Nevertheless, to date, research on the mechanism(s) influencing cross-feeding among eukaryotes and its impacts on cellular heterogeneity and exogenous plasmid retention rate is limited.

Through studying plasmid persistence within a population of plasmid-complemented *ura3ΔO Saccharomyces cerevisiae* cells, we found that pyrimidine cross-feeding between individuals can greatly compromise auxotrophic selection pressure and promote cell-to-cell variation in recombinant productivity, both of which can be harnessed with appropriate genetic manipulation strategies. In addition, we pivot the findings to develop recombinant strains that have high plasmid retention rate even when cultivated under non-selective environments, as well as proof-of-concept designs of yeast-yeast and yeast-*Escherichia coli* co-culture systems that can be further optimised for practical applications. While the study focuses primarily on microbial fermentation, we envision the knowledge gathered in this research not only benefits bioengineering but also extends into other biotechnological applications, such as the treatment of certain biofilms and cancer cell types.

## Results and discussion

### A significant proportion of plasmid-free cells can exist under auxotrophic selection in an unexpectedly predictable manner

From a canonical perspective, it is only feasible for *ura3ΔO* yeast auxotrophs to grow in a pyrimidine-dropout selection environment once the cells have been successfully transformed with and carry the complementary *URA3*-containing plasmid(s). This, nevertheless, is not completely true.

Using flow cytometry, we initially observed that living BY_mTU3 yeast cells, transformed with pU3-GAL1_mT2 episomal plasmids, within single colonies growing on SC-Ura selective agar plate are already highly heterogeneous with respect to fluorescent protein (FP) recombinant expression (Fig. 1A), wherein two distinct subpopulations could be identified: one with significant FP signal (i.e. FP signal-positive, hereafter referred to as "FP+ve") and one whose FP signal is indistinguishable from the autofluorescence of non-induced cells (i.e. FP signal-negative, hereafter abbreviated as "FP−ve"). In addition, single-cell fluorescence intensity among FP+ve individuals is also broadly variable, with larger cells emitting higher FP intensity. Similar bimodal expression at the population level was also generated when BY_mTU3 was cultivated in SC-Ura liquid media (Fig. 1A). Notably, such bimodal recombinant expression was observable in yeast strains that have an FP reporter gene anchored on high-copy plasmids with an auxotrophic selection marker (Fig. S1A–C) but not in strains which have the FP gene chromosomally integrated (Fig. S1D), which aligned with previous reports[27,28]. Bimodal recombinant expression from high-copy plasmids was independent of the promoter, choice of FP reporter, or the auxotrophic selection marker used.

Further examination with fluorescence-activated cell sorting (FACS) verified that the FP−ve cells within the BY_mTU3 population are indeed plasmid-free as just one colony resulted from plating a FACS-separated "mTur2 −ve" sample (98.9% of the population is FP−ve) on SC-Ura agar (Fig. 2A; Fig. S2). This eliminated any possibility that the FP−ve subpopulation was plasmid-bearing but, for example, had its FP expression silenced by unknown genetic-/ non-genetic causes (as in the case of clone BY_mTU3#18 in Fig. S3, discussed below). The much higher single-cell median fluorescence intensity (MFI) along with larger variance (rCV) among FP+ve, high-copy plasmid-carrying yeast cells over strains where the FP gene is integrated into the chromosome also

infers that, besides plasmid loss, another major contributor of cell-to-cell variation in recombinant protein productivity is gene-copy-number variation among recombinant cells (Fig. S1D), as single-cell FP gene dosage is proportional to its fluorescent intensity read-out[29]. This also explained why high-copy 2μ plasmid expression systems are generally more heterogeneous than low-copy CEN6/ARS4 plasmid and chromosomally integrated systems at the population level[28].

When we patch-plated BYmTU3 colonies on SC-Ura and SC agars and analysed colony compositions 2 weeks after (Fig. 1A), 62/65 colonies on SC-Ura plates showed the same bimodal FP expression pattern consisting of 44.2 ± 3.7% of FP+ve cells. In comparison, the median plasmid maintenance rate of colonies that grew on non-selective SC plates was much inferior (11.9 ± 3.3% of FP+ve cells), suggesting plasmid segregational instability[30,31] alone is not sufficient to explain the proportion of plasmid-free cells presents in selective media. The three rare anomalies (BY_mTU3#4, #12 and #18) on SC-Ura plates were assumed to sustain spontaneous mutation(s), causing their respective FP and FSC (forward scatter, proportional to cell size) profiles to differ (Fig. S3).

To better understand how the relative abundance of plasmid-carrying cells over plasmid-free cells (Equiv. FP+ve: FP−ve cell ratio) dynamically changes in a selective environment over a conventional batch cultivation process, eight BY_mTU3 clones were propagated in SC-Ura liquid media and their FP expression profiles were monitored via flow cytometry (Fig. 1A). Strikingly, temporal development of population FP expression profile over 72 h of batch fermentation was almost identical across the yeast cultures, albeit the small variations during early log phase (8 h) that coincided with the slightly higher standard deviation in population optical density at this growth stage. In other words, the only observable differences between the isogenic yeast populations were the duration of the lag phase, and the initiation timing of recombinant protein expression, and such phenotypic differences are believed to be underlined by the stochastic variation in gene expression among isogenic cells that ultimately dictates their likelihood and efficiency in escaping the lag phase upon environmental shift to eventually resume growth[32].

When we repeated in a 32-h batch fermentation with more sampling points in between and cross-referenced the growth curve to the FP expression profile of the BY_mTU3 cultures (Fig. 1B), we found that the proportion of FP+ve cells within the population increased in a temporal-dependent manner after recombinant expression was galactose-induced (0–12 h) up until a maximum was reached (~12 h), after which the %FP+ve cells decreased gradually (12–24 h) until this plateaued (24–32 h) as the population entered stationary phase and cell budding ceased. Of note, the viability of the cell cultures remained high by the end of the fermentation, with ~99% of the population having intact cell membranes and thereby inferred to be metabolically capable.

Further investigation revealed the plasmid-free cells within the BY_mTU3 population were actively growing despite being under selection pressure. Particularly, the ratio of budding cells (i.e. doublets) over singlets during the exponential phase was comparable between the FP−ve and FP+ve subpopulations (9%/16.8% vs. 26%/48.2%) (Fig. 1B). In addition, the mitochondrial membrane potential of the FACS-separated FP−ve and FP+ve subpopulations were found to be very similar, and both were comparable to that of the unsorted mixed population (FACS Ctrl) (Fig. S4), which indicates the growing FP−ve and FP+ve cells had the same vitality. This finding essentially explains why the proportion of plasmid-free cells within BY_mTU3 colonies and liquid cultures was much higher than the published segregation rates of yeast episomal plasmids (typically < 7%)[6,30].

### The growth of plasmid-free cells under auxotrophic selection pressure depends on co-existing plasmid-carrying cells

The presence of metabolically active plasmid-free cells co-existing along with complementary plasmid-carrying cells in a selective

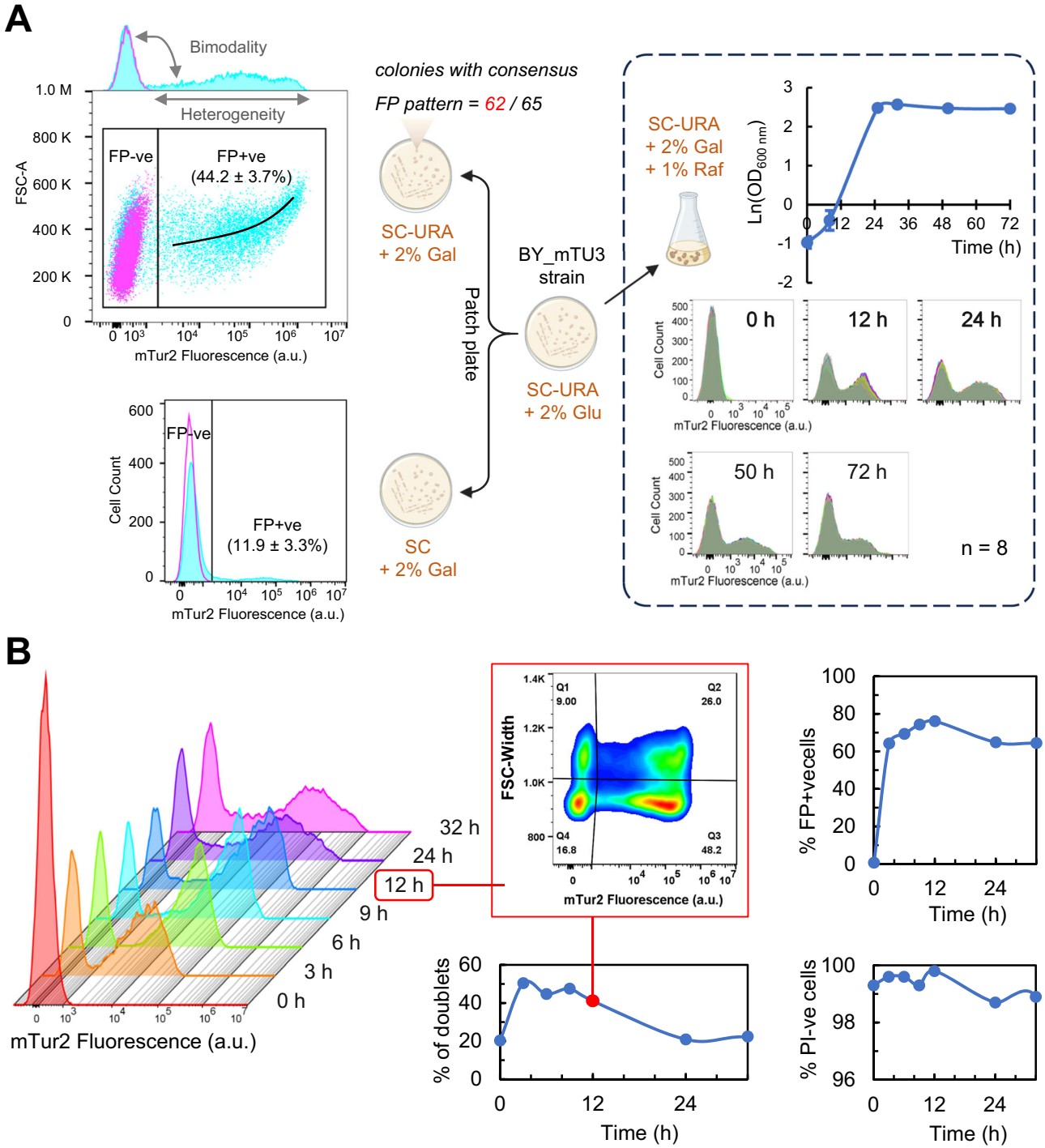

**Fig. 1 | Cell-to-cell heterogeneity in recombinant protein expression within a high-copy plasmid-carrying yeast population maintained under auxotrophic selection. A** Transformed yeast cells were initially selected on SC-Ura agar plates supplemented with 2% glucose. Subsequently, flow cytometry was used to assess the bimodal mTur2 expression profile of BY_mTU3 strain when cultivated either on SC-Ura selective agar plate containing 2% galactose (*n* = 62 colonies), on SC non-selective agar plate containing 2% galactose (*n* = 62 colonies), or in SC-Ura liquid media containing 2% galactose + 1% raffinose (*n* = 8 biologically independent cell cultures propagated from 8 separate colonies). Totally, 62/65 clones isolated on SC-Ura agar plate showed highly similar mTur2 expression profiles when induced with galactose (cyan), of which the peak of the subpopulation with "background" mTur2 signal overlapped with non-induced cells (magenta, FP−ve). The average percentage of mTur2-expressing yeast cells within the colonies (%FP+ve) on SC-Ura and SC plates, respectively, are reported as "median ± s.d.". The population mTur2 profiles in liquid media also closely overlapped and showed dynamic change with respect to the population growth curve over 72 h of batch mode cultivation in shake flasks (growth data points were presented as mean values ± s.d.). **B** The temporal changes in population mTur2 expression (left), the proportion of mTur2-expressing subpopulation (%FP+ve cells, top right), cell with high membrane integrity (percentage of propidium iodide-negative (%PI−ve) cells, bottom right), and population proliferation status (percentage of doublets, middle, with data callout for t = 12 h, wherein Q1 and Q2 represent doublets that are non-mTur2 expressing and mTur2-expressing, respectively) within a liquid culture across a 32-h batch mode fermentation. The experiment was conducted in two bio-replicates (*n* = 2 independent cell cultures propagated from different clones), with no significant differences observed statistically between the two (see Source Data file). Cartoon graphics were created with BioRender.com.

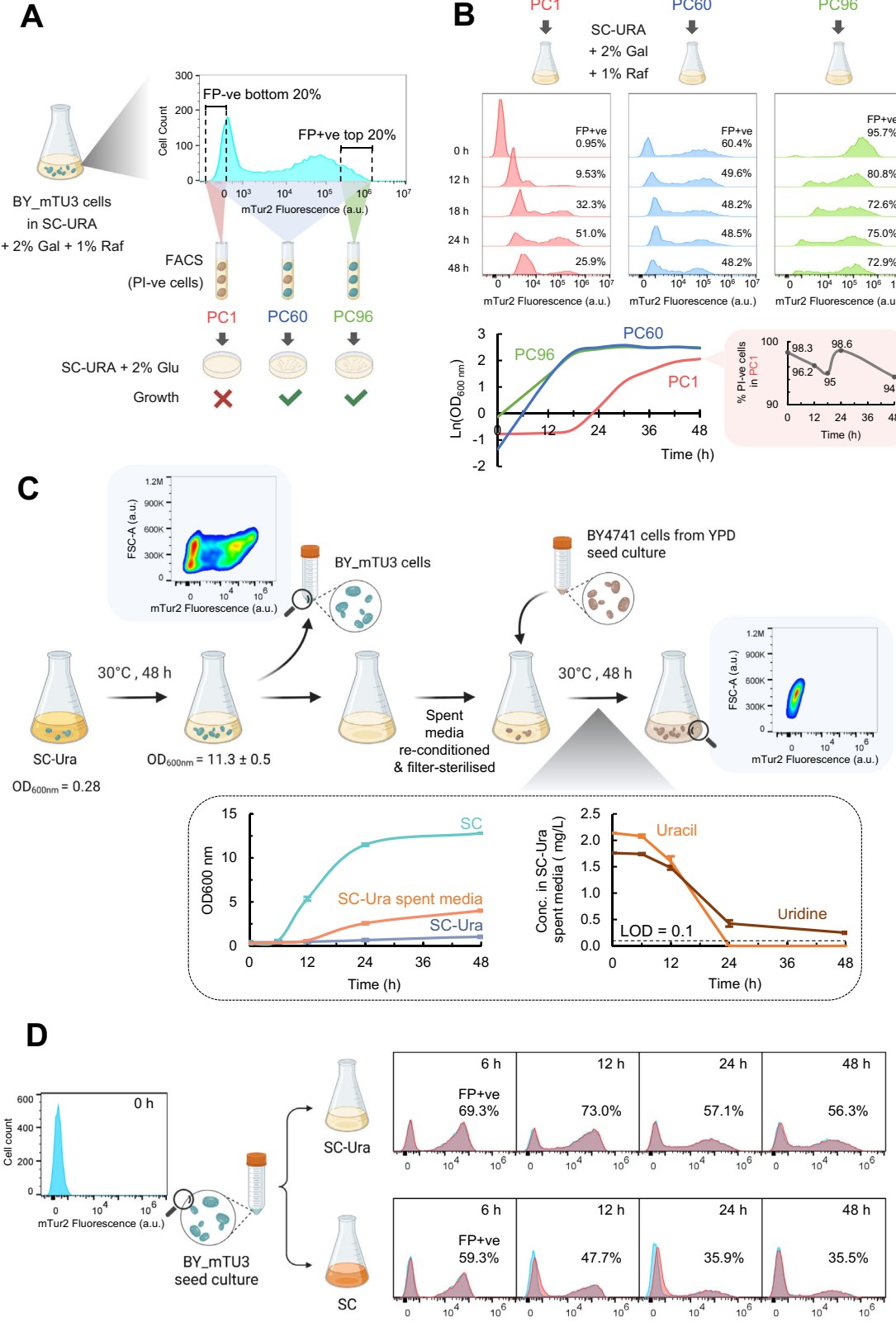

medium has been previously reported. Nevertheless, the underlying biology of such phenomenon is still a subject of debate, with three major hypotheses proposed. Firstly, by placing the auxotrophic selection marker gene on a plasmid, the plasmid-carrying cells could potentially over-synthesise the complementary metabolite(s) (CMs), wherein the residual intracellular excess could be inherited by plasmid-free daughter cells and allow them to undergo several divisions within

a selective environment after the mitotic event[33–37]. However, when Shou and colleagues cultivated *lys2Δ* and *ade8Δ S. cerevisiae* strains firstly as monocultures in synthetic complete (SC) media and subsequently co-cultivated them into SC-Lys-Ade media, they reported both strains quickly died after an initial, transient growth period of ~10 h using the stored metabolites[22]. Hence, the growth of plasmid-free cells facilitated solely by intracellularly inherited CMs under auxotrophic

**Fig. 2 | Plasmid-free auxotrophs depend on the cross-feeding of complementary metabolites from plasmid-carrying prototrophs for growth in a selective environment. A** Cell sorting strategy employed to separate the subpopulations expressed (FP+ve) and not expressed (FP−ve) fluorescent proteins within a BY_mTU3 culture exponentially growing in SC-Ura selective medium, as well as their respective culturability on SC-Ura + 2% glucose selective agar plates after fluorescence-activated cell sorting (FACS, wherein only cells that were negative for propidium iodide staining (PI−ve) were selected) (See Fig. S2). **B** mTur2 expression profiles (upper) and growth curves (lower left) of three BY_mTU3 populations with different initial ratios of plasmid-carrying: plasmid-free cells when being cultivated in SC-Ura media (containing 2% galactose + 1% raffinose) under batch mode (n = 1); The percentage of viable, PI-negative cells within the PC1 culture at each sampled time point was also presented in lower right. Cell count displayed for each flow cytometry data panel = 4000. **C** Population growth of

BY4741 cells *(ura3Δ0)* in SC non-selective media, SC-Ura selective media, as well as "SC-Ura" spent media collected from a 48 h-old BY_mTU3 cell culture and subsequently reconditioned and filter-sterilised (left), along with the consumption of pyrimidines (uracil and uridine) by BY4741 cells during their cultivation in the "SC-Ura" spent media (right) over 48 h of batch mode fermentation (n = 3 independent cell cultures, data points in line graphs were presented as mean values ± s.d.). Flow cytometry insets verified that all FP+ve, pyrimidine-producing cells were removed from the "SC-Ura" spent media collected at the end of the BY_mTU3 culture, as proven by their absence at the end of the subsequent BY4741 cell culture in the reconditioned "SC-Ura" spent media. **D** mTur2 expression profiles of the BY_mTU3 strain when cultivated in either SC or SC-Ura media using 2% galactose and 1% raffinose as carbon sources under batch mode (n = 2 independent cell cultures propagated from different clones). Cartoon graphics were created with BioRender.com.

selection is not sustainable. Alternatively, it was proposed that the over-synthesised CMs can either be released upon cell lysis[38–40] or be excreted by plasmid-carrying cells into the selective medium and then scavenged by plasmid-free cells for growth[30,39,41–45]. While the release of CMs from extensive lysis of plasmid-carrying cells could support the growth of co-existing auxotrophs in the culture to some extent[22,46] since Fig. 1B clearly shows plasmid-free (FP−ve) cells within BY_mTU3 population proliferated in SC-Ura medium while cell death events are minimal, cryptic growth on CMs released from cell lysis should not be the main mechanism supporting plasmid-free cell growth in selective environments either.

Collectively, this leaves the theory of CMs cross-feeding to be investigated. As previously mentioned, bacteria that are auxotrophic for essential biomolecules such as amino acids or nucleobases are capable of effectively exchanging such metabolites with co-existing cells to overcome their own metabolic deficiencies and proliferate[15–19]. In yeasts, however, historically, it has proven to be more challenging to establish a complementary cross-feeding system that can support the growth of auxotrophs unless the co-residing CM-producing yeast cells are genetically modified CM-overproducers[22–24]. In fact, it was only until very recently that the metabolite-sharing activities of yeasts, both in situ and cross-generational[5,21,47], were experimentally proven. Nevertheless, within a recombinant yeast population wherein plasmids are maintained under auxotrophic selection, whether the extent of metabolite exchange can occur at significant magnitude to allow plasmid-free auxotrophs to overcome selection pressure and proliferate at scale has so far only been investigated via mathematical modelling by Sardonini and DiBiasio[43] and has yet to be experimentally substantiated. One elegant model created by Campbell and colleagues exploited plasmid segregational loss to generate a self-established, metabolically cooperating yeast community (SeMeCo)[6]. The model implies yeast possesses a natural capacity for metabolite exchange at life-supporting quantities even without being converted into metabolite overproducers through feedback-resistant mutations. Nevertheless, the uracil, leucine, histidine and methionine auxotrophs isolated from an established SeMeCo were still unable to complement each other's deficiencies upon co-culturing[6], either in pairwise combinations or as mixed ratios of all four in corresponding selective media, implying the growth of auxotrophs in a proliferating SeMeCo exhibits absolute correlation to the co-presence of prototrophs which they can resource complementary metabolites from. Because of the reason stated above, the SeMeCo always maintains a considerable proportion of prototrophs when it proliferates under a selective environment[6]. Consequently, as prototroph plasmid-shedding, intracellular CMs inheritance, cell lysis and cross-feeding can all contribute to the increase in auxotrophic cell counts in SeMeCo, it is difficult to distinguish the extent of contribution of CMs cross-feeding to auxotrophic cell growth within this synthetic community.

More recently, Aulakh and colleagues conducted high throughput screening through the Yeast Knockout Collection (YKO) and identified

49 auxotrophic pairs out of 1891 tested (2.6%) that were able to form syntrophic co-cultures in selective media[21,48]. Most of the successful pairs were between amino acid auxotrophs; however, regarding pyrimidine auxotrophs, only *ura1Δ* can form syntrophic co-cultures, and only with *met13Δ* or *aro2Δ* out of 91 growth pairs tested[21]. Nevertheless, the mechanism(s) behind the success of the small collection of syntrophic pairs remains unknown, and the impact of syntrophic cross-feeding on plasmid retention was also not examined in these strains.

In a mixed culture of prototrophs and its derived auxotrophs, a major premise to favour cross-feeding over intracellular CMs inheritance hypotheses is that the growth of auxotrophic plasmid-free cells within the population must be dependent on the co-presence of prototrophic plasmid-carrying cells. To verify this point, we used FACS to generate three BY_mTU3 cultures with different initial FP+ve: FP−ve cell ratios of ~1% (PC1), ~60% (PC60), and ~96% (PC96), respectively, and monitor the temporal change in their population FP expression pattern upon re-cultivation in selective media (Fig. 2B). We found that when the fraction of plasmid-carrying cells within the initial seed culture was abundant (PC60 and PC96), the FP+ve: FP−ve cell ratio gradually decreased over time, with the %FP+ve cells reduced from ~60% to ~48% in PC60 and from ~96% to ~73% in PC96 after 48 h batch fermentation. While, in the PC1 culture wherein there was only ~1% of FP+ve, plasmid-carrying cells at 0 h, a distinctive long lag phase was observed (0–18 h) before population growth was detected, which was not manifested in either PC60 or PC96 cultures. Coincidently, population growth in PC1 was contributed predominantly by plasmid-carrying cells from 0 to 24 h as they increased fractionally from ~1% to ~51% during this period before being outgrown by plasmid-free cells at the later growth phase (24–48 h). Seemingly, plasmid-free cell proliferation was stalled in the early hours after inoculation into SC-Ura media, and since cell lysis was minimal throughout the batch fermentation, their subsequent growth showed a strong dependency on the relative abundance of co-existing plasmid-carrying cells. Interestingly, when the PC1, PC60, and PC96 yeast cultures accumulated more generations during a three-stage cultivation process, their FP+ve: FP−ve cell ratios eventually converged to become identical (Fig. S5).

The findings collectively suggest that the plasmid-free: plasmid-carrying cell ratio of the BY_mTU3 population in SC-Ura media is governed by a negative frequency-dependent selection, potentially a case that fits the Black Queen Hypothesis[49]. Under this scenario, the growth of plasmid-free cells in an auxotrophic selective environment is supported by CMs excreted from plasmid-carrying cells, in which case, the abundance of the former is dictated by the frequency and CMs productivity of the latter within the mixed population. Such hypothesis was further consolidated here when the filter-sterilised, reconditioned spent SC-Ura media collected from a 48-h batch fermentation of BY_mTU3 was shown to effectively support the growth of BY4741 *(ura3Δ0)* pyrimidine auxotrophs (Fig. 2C). Additionally, both uracil and uridine were detected in the SC-Ura fermentation media during the BY_mTU3 growth phase, and both were subsequently consumed

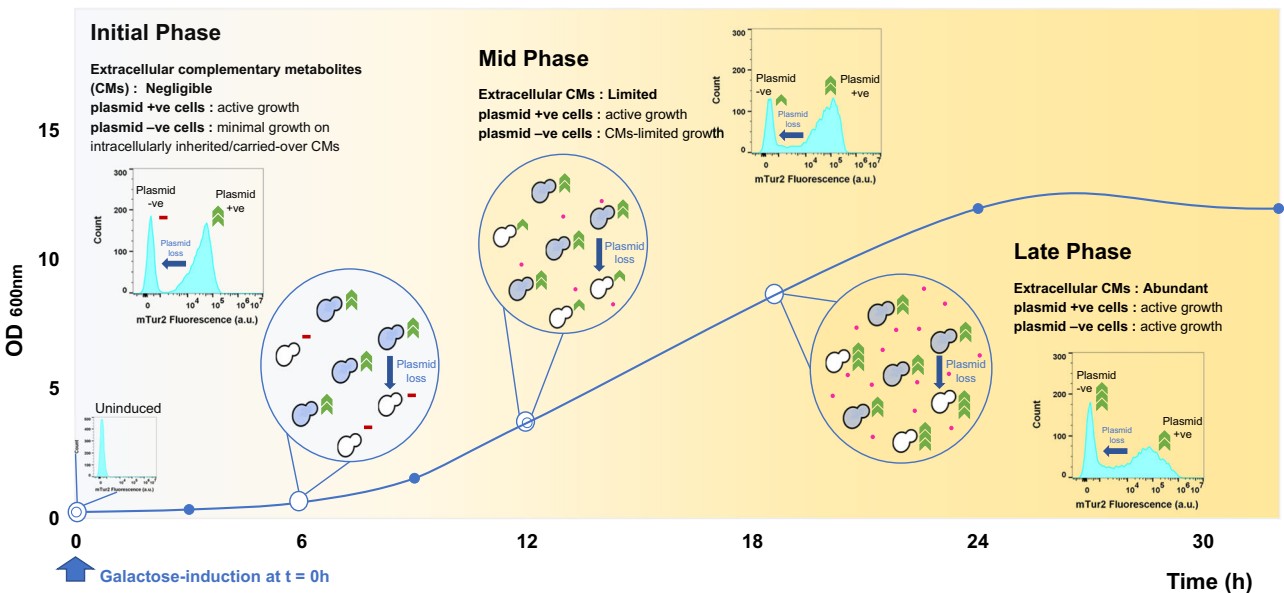

**Fig. 3 | The relative growth rate of plasmid-free auxotrophs and plasmid-carrying prototrophs is a function of progressive complementary metabolite (CM) accumulation within selective media.** Blue-shaded cells represent plasmid-carrying cells. White cells represent plasmid-free cells. Magenta dots represent CM molecules excreted into the selective media by plasmid-carrying cells. Blue arrows represent plasmid-free cells generated through plasmid segregational loss. Red bars represent minimal cell growth supported by intracellular CMs that are either inherited from mother cells or carried over from historical non-selective cell growth prior to the application of auxotrophic selection pressure. Green arrowheads represent active cell growth, with the number of arrowheads representing the relative growth rate between the two subpopulations (more arrowheads = faster growth rate). The growth of plasmid-carrying cells is proposed to follow Monod kinetics with a rate-limiting substrate S, while the growth of plasmid-free cells can be described by a dual Monod form with growth being a function of both substrate S and complementary metabolite CM[43]. Under conditions where CM is not a growth-limiting factor, it is proposed that plasmid-free cells grow faster than plasmid-carrying cells since they do not incur plasmid-related metabolic costs[50]. Flow cytometry insets are the mTur2 expression profiles of a BY_mTU3 population at various growth stages in SC-Ura media in batch fermentation (the complete dataset is presented in Fig. 1B), collectively elaborate the change in the ratio of plasmid-carrying cells over plasmid-free cells within the culture as a function of the accumulation of pyrimidines excreted by plasmid-carrying cells into the media over time. Cartoon graphics were created with BioRender.com.

when spent media was used to support the growth of a fresh culture of BY4741 auxotrophs (Fig. 2C). Further, the proportion of plasmid-free cells within the BY_mTU3 mixed population was enriched when uracil was exogenously supplied (Fig. 2D). Specifically, when an inoculum of BY_mTU3 cells was separated into SC and SC-Ura media and incubated over 48-h batch fermentation, the %FP+ve cells within the SC-Ura culture were higher than that of the SC culture throughout and significantly at 48 h (~56% vs. ~36%, respectively). In addition, when BY4741 cells were grown in non-selective SC media before being transferred into fresh SC-Ura selective media, the subculture density only incrementally increased (Fig. 2C). Hence, although intracellularly inherited CMs do allow auxotrophs to proliferate under selection pressure as suggested by previous reports, population growth on such source of CMs is neither sustainable nor significant at a bioprocess scale.

The results above essentially validated CMs cross-feeding is indeed the underlying mechanism that supports plasmid-free cells to escape auxotrophic selection, and align well with the mathematical model previously proposed by Sardonini and DiBiasio[43]. A schematic depicting the relative growth rate of plasmid-free auxotrophs and plasmid-carrying prototrophs in selective environment as a function of CMs availability in the exometabolome throughout a batch cultivation process is presented in Fig. 3: In the case of BY_mTU3, initially when the population containing a mixture of plasmid-free *ura3Δ0* auxotrophs and plasmid-carrying prototrophs was inoculated into fresh SC-Ura media lacking any pyrimidine, the population growth was dominated by plasmid-carrying cells. As the fermentation proceeded, pyrimidines were excreted by plasmid-carrying producers and made available to be taken up by all cells in the culture for growth. When the level of pyrimidines in the media built up and became abundant at the later

growth stages, plasmid-free cells were no longer pyrimidine-limited and began to outgrow plasmid-carrying cells since they do not incur any plasmid-associated metabolic costs (i.e. plasmid maintenance, plasmid replication, and recombinant gene expressions[50]).

The extensive excretion of pyrimidines from plasmid-carrying cells is thought to be driven by their intracellular overproduction of pyrimidines as they experienced *URA3* selection gene "overdose", which subsequently triggers directed overflow metabolism to maintain intracellular pyrimidine homoeostasis[51]. Given pyrimidine overflow excretion is likely an ATP-dependent mechanism instead of being accomplished through passive leakage[52], and considering pyrimidines are essential for the growth of plasmid-carrying cells themselves, we postulate that plasmid-carrying cells release these over-produced pyrimidines primarily for self-preservation interests rather than communal purposes or in an attempt to engage in synergistic metabolic interactions with the plasmid-free auxotrophs. Indeed, intracellular over-accumulation of deoxyuridine triphosphate (dUTP) was shown to be lethal since it can lead to extensive uracil-substituted DNA[53].

## Constraining synthesis and/or import of complementary metabolites represses plasmid-free cell growth

The fact that CMs cross-feeding can compromise auxotrophic selection effectiveness has great implications. Specifically, multi-copy plasmids maintained via auxotrophic marker genes are still the preferred yeast recombinant expression platforms over antibiotic selection plasmids and chromosomal integration systems under many circumstances, for instance, when high single-cell recombinant productivity is favoured, when process scale-up is desired, and when regulations concerning human safe consumption need to be met such as those in biopharmaceutical and food industries. Therefore,

engineering strategies to mitigate the impact of CMs cross-feeding is highly desired and necessary when such a biological phenomenon is deemed unfavourable.

In *URA3* plasmid-complemented *S. cerevisiae* strains with either *ura3Δ fur1Δ* mutations[54] (derivatives of wild type strain FL100) or *ura3Δ fur1Δ urk1Δ* mutations[55] (derivatives of YM603, which is congeneric with wild type strain S288C[56]), it was previously reported that plasmid-free cells were effectively eliminated within the population even when non-selective media was used, since these strains lack functional FUR1p (uracil phosphoribosyltransferase) and URK1p (uridine kinase) to convert the salvaged pyrimidines into uridine monophosphate (UMP) (Fig. 4A). Of note, these strains were constructed by exposing *URA3* plasmid-complemented *ura3Δ* yeast cells to a series of fluor-opyrimidines to isolate spontaneous mutants of *FUR1* and *URK1* genes[54,55]. Regrettably, these "auto-selection" genotypes could not be generated using BY_mTU3 (BY4741 background) as a base strain, despite attempts via different homologous recombination strategies, including hygromycin resistance gene (HphMX4) or auxotrophic selection markers (LEU2 and HIS3) for gene disruption. This may be due to differences in the genetic background of target yeast as loss-of-function mutations have a high dependency on genetic background; that is, deletion of the same metabolic gene in a different strain background could lead to an entirely different transcriptional response[57,58]. Furthermore, given the *fur1Δ* mutant was previously reported to accrue high fitness cost[41], it is certainly possible that the viability of *fur1Δ* and *fur1Δurk1Δ* mutants are dependent on strain background, and that in BY4741 (*ura3Δ*) the complementary pESC-URA3 plasmids could not sufficiently rescue these genotypes from the lethal effect of the *ura3Δ fur1Δ* double mutations.

As alternative strategies, we proposed to impair the capability of plasmid-free cells to uptake extracellular pyrimidines by knocking out respective genes encoding either the high-affinity uracil permease (FUR4p) or uridine permease (FUI1p) (Fig. 4A); or reduce the intracellular synthesis of these metabolites within plasmid-carrying cells such that pyrimidine overflow and the consequential excretion could be minimised in situ, which can be achieved through engineering the complementary selection marker (i.e. using the defective URA3d or URA3dCL1 in place of the fully functional URA3) (Fig. 4A). Under both strategies, the extent of CMs cross-feeding and thereby the growth of plasmid-free cells in selective media would expect to be suppressed, leading to a more homogeneous cell culture with higher proportion of FP+ve, plasmid-carrying cells (Fig. 4B).

We compared the frequency of plasmid-carrying cells in SC-Ura of BY_mTU3 to yeast strains derived from the abovementioned genetic engineering strategies. We found that while the BY*fui1Δ*_mTU3 strain showed little improvement in plasmid retention rate over BY_mTU3 after 48-h batch fermentation (%FP+ve $_{48-12 h}$ = 10.7% for BY*fui1Δ*_mTU3 and = 13.9% for BY_mTU3), the plasmid retention rate of BY*fur4Δ*_mTU3 cultures were on the other hand greatly improved (%FP+ve $_{48-12 h}$ = 3.2%) (Fig. 4C). The fact that FUR4p was found to be the more prominent pyrimidine importer aligns well with results presented in Fig. 2C that uracil is the preferred cross-feeding pyrimidine over uridine, as it is both excreted more into the exometabolome by the plasmid-carrying producers as well as being taken up at a higher rate by plasmid-free consumers.

Moreover, by using defective selection markers to modulate pyrimidine synthesis rate within plasmid-carrying producers, minimal live plasmid-free cells were observed within both the BY_mTU3d and BY_mTU3dCL1 cultures as they exhibited unimodal mTur2 expression profiles, with plasmid-carrying cells making up ~98% of the former and 100% of the latter at 48 h (Fig. 4C). Similarly, in the LEU2 auxotrophic selection system, replacing LEU2 with LEU2d defective selection marker also resulted in drastic reduction in %FP−ve, plasmid-free cells within SC-Leu liquid culture (Fig. S6), which suggests CMs cross-feeding is a universal issue encountered by all auxotrophic selection

systems where the CMs can be excreted. Consistently, in non-selective SC media wherein exogenous uracil was available, plasmid-free auxotrophs derived from BY_mTU3d and BY_mTU3dCL1 cells once again became proliferative as they were capable of uptake uracil from the media via their pyrimidine permeases (Fig. S7). Together, the results indicate both the BY_mTU3d and BY_mTU3dCL1 strains did not over-synthesise pyrimidines and thereby minimally excreted and available for the plasmid-free cells in SC-Ura media to scavenge. Without the availability of CMs to cross-feed, the presence of plasmid-free auxotrophs in selective media will never be significant even where auxotrophs continue to be generated during population growth as plasmid segregational loss in high-copy 2μ plasmids cannot be prevented. The results also reconsolidated the notion that CMs over-synthesised within plasmid-carrying yeast cells were excreted through an active, ATP-dependent process rather than through passive leakage[52], as any passive leakage of pyrimidines would have facilitated the growth of plasmid-free cells within the BY_mTU3d and BY_mTU3dCL1 cultures in SC-Ura media, which was not the case (Fig. 4C).

In addition to the complete plasmid-retention rate in SC-Ura media, BY_mTU3dCL1 cells also have higher single-cell productivity (MFI) and lower cell-to-cell heterogeneity (rCV) regarding recombinant protein expression compared to BY_mTU3 and BY_mTU3d. Specifically, at 24 h post-induction, the MFI of BY_mTU3dCL1 cells is >8-fold higher than that of BY_mTU3 cells (which becomes >13-fold after 48 h post-induction), while having an rCV value less than one-fourth to that of BY_mTU3 (Fig. 4D). This superior phenotype implies BY_mTU3dCL1 cells carry higher plasmid copy number (PCN)[29], which can be explained by the lower complementation efficiency of the URA3dCL1 selection marker, such that the minimal PCN required per cell for BY_mTU3dCL1 to overcome the auxotrophic selection pressure and proliferate is higher than BY_mTU3 and BY_mTU3d. While previous studies reported that yeast strains carrying URA3d or LEU2d selection markers generally have increased population-average PCN compared to strains carrying URA3 or LEU2 selection markers[40,59,60], Fig. 4C and Fig. S6 illustrate that while the plasmid-carrying URA3d/LEU2d cells may indeed have higher average PCN compared to plasmid-carrying URA3 /LEU2 cells, the major contributing factor to the increase in population-average PCN value however can be rationalised by the reduction in the frequency of plasmid-free cell within the URA3d/LEU2d populations. This is statistically supported by the fact that the MFI of BY_mTU3d cells is only 1.7–2.4 folds higher than that of BY_mTU3 (Fig. 4D). Additionally, BY_mTU3dCL1 manifested slightly subpar growth fitness compared to BY_mTU3 and BY_mTU3d in both selective and non-selective media (Fig. 4D and Fig. S7), and batch cultures of BY_mTU3dCL1 in SC-Ura media also accrued more cells with compromised plasma membrane after 48 h (Fig. 4C).

Previous studies have shown that haploid *S. cerevisiae* harbouring high-copy plasmids with fully functional auxotrophic selection marker grew slower than their plasmid-free counterparts due to plasmid-related metabolic costs[50], even when non-selective media was used[61], and such fitness cost is expected to exacerbate in strains with highly defective selection markers like BY_mTU3dCL1 wherein single-cell PCN can increase many folds. Moreover, the amount of recombinant protein expressed is also an expensive metabolic cost, and once it makes up 30% of the host's internal proteome, intracellular protein toxicity was shown to become significant[62]. Nevertheless, the increase in population ensemble recombinant productivity in both BY_mTU3d and BY_mTU3dCL1 is significant both statistically and visually, with their recombinant protein expression level per O.D.$_{600 nm}$ being 2.1 and 3.6-fold higher, respectively, than BY_mTU3 culture at 24 h post-induction when cultivated in SC-Ura media (Fig. 4D).

In summary, in an auxotrophic selective environment, CMs excreted by plasmid-carrying prototrophs into the exometabolome would not only facilitate plasmid-free cells to escape selection pressure but also promote metabolic heterogeneity among the plasmid-

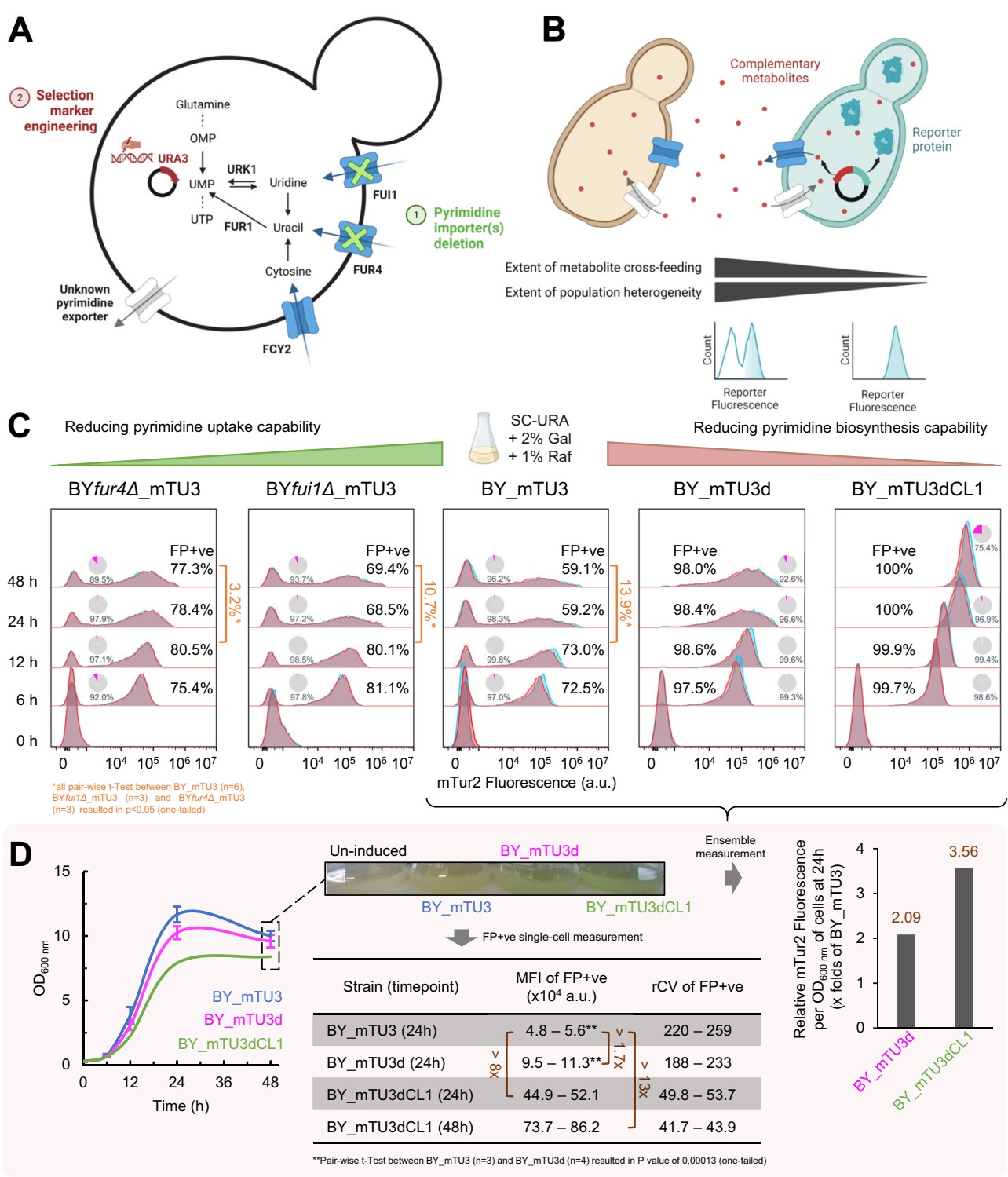

carrying subpopulation itself. Specifically, both prototrophs and auxotrophs prefer to uptake CMs, when available, from the external environment over self-synthesis[6,63]. In fact, Campbell and colleagues found that when CMs are externally available, prototrophic yeasts would consume the required CMs at rates and quantities comparable to the corresponding auxotrophs[6]. As such, the availability of CMs in the exometabolome will lower the minimum PCN threshold required for plasmid-carrying cells to survive in minimal media, thereby allowing cells with lower PCN to proliferate. The interplay and modulation of intracellular PCN by extracellular CM abundance further reaffirm the notion that both exo- and intra-metabolomes are crucial parts of

the metabolism of individual cells, and the composition of the exometabolome can influence the survival of cellular residents within it under stressful conditions, in this case, pyrimidine limitation being experienced by plasmid-free auxotrophs derived from BY_mTU3d and BY_mTU3dCL1 plasmid-carrying cells.

Furthermore, compared to bacteria, the larger-sized yeast cells have higher energetic and metabolic demands for both cell maintenance and growth[64], while their smaller surface area-to-volume ratio causes them to be less efficient in nutrient acquisition and scavenging from the surroundings due to diffusion limitations of nutrient transport[65-67]. Hence, extending from the experimental data

**Fig. 4 | Repressing complementary metabolite(s) cross-feeding within recombinant yeast population can mitigate both plasmid-free cell growth under auxotrophic selection pressure as well as population heterogeneity in recombinant expression. A** Simplified schematic of the pyrimidine de novo biosynthesis and salvage pathways in *S. cerevisiae*, as well as the potential genetic engineering targets to reduce the cell's capability to cross-feed pyrimidines— namely 1. pyrimidine importer(s) deletion and 2. selection marker engineering. **B** Illustration on the hypothetical correlation between complementary metabolite (red dot) cross-feeding and cell-to-cell heterogeneity in recombinant productivity (cyan reporter protein) within a recombinant yeast population wherein the transformed plasmids were maintained via auxotrophic selection. **C** mTur2 expression profiles of BY_mTU3, BY*fui1Δ*_mTU3, BY*fur4Δ*_mTU3, BY_mTU3d, and BY_mTU3dCL1 populations when being cultivated in SC-Ura media (containing 2% galactose + 1% raffinose) for 48 h under batch mode. Flow cytometry data of two independent cell cultures were presented ($n = 2$), with pie chart insets indicating the proportion of cells that were negative for propidium iodide staining (%PI−ve, intact cell membrane) in grey and those positive for propidium iodide staining (%PI+ve, compromised cell membrane) in magenta. Pair-wise *t*-tests between BY_mTU3 ($n = 6$), BY*fui1Δ*_mTU3 ($n = 3$), and BY*fur4Δ*_mTU3 ($n = 3$) strains were conducted to verify the statistical significance in the difference in their plasmid retention rate (measured as the difference in the percentage of mTur2-expressing cells (%FP+ve) within the population at $t = 12$ h and $t = 48$ h), which yielded one-tailed *P* values of: 0.00021 between BY_mTU3 and BY*fur4Δ*_mTU3, 0.041 between BY_mTU3 and BY*fui1Δ*_mTU3, and 0.00014 between BY*fur4Δ*_mTU3 and BY*fui1Δ*_mTU3. **D** Growth curves of BY_mTU3 ($n = 3$), BY_mTU3d ($n = 4$), and BY_mTU3dCL1 ($n = 2$) populations (left, growth data points were presented as mean values ± s.d.), the cultures' visual appearance at the end of the fermentation (middle top), as well as the population-average (right, data included in Table S7) and single-cell (middle bottom) measurements of their mTur2 recombinant expression at 24 h and 48 h post-induction in SC-Ura media. The median fluorescent intensity (MFI) and robust coefficient of variance (rCV) of FP+ve cells within the cultures are presented as a range given by the highest and lowest values measured across bio-replicates via flow cytometry. *t*-test comparing the MFI of FP+ve cells within the BY_mTU3 and BY_mTU3d cultures yielded a *P* value of 0.00013 (one-tailed). **C, D:** $n =$ independent cell cultures propagated from separate clones. **D** Error bar was not included for the growth curve of BY_mTU3dCL1 due to limited sample size ($n = 2$); reproducibility across its two bio-replicates is enclosed in the Source Data file. Cartoon graphics were created with BioRender.com.

observed here, the previously reported challenges in establishing viable co-cultures between two complementary auxotrophic *S. cerevisiae* strains may be due to the gene dosage engineered into the yeast strains being too low and therefore, CMs production was insufficient to support complementation. That is, the strains employed in these studies generally contain only one to a few copies of the chromosomally-integrated selection marker gene(s) and thus require the addition of metabolic feedback resistance mutation(s) to trigger directed overflow metabolism that would result in the cross-feeding of CMs at life-supporting quantities[22–24,68]. This conclusion is consistent with the very few *S. cerevisiae* auxotrophs that can form syntrophic growth without feedback-resistant mutations; Aulakh and colleagues found that cultivation success of these syntrophic cultures was sensitive to population ratios upon initial inoculation[21] since different auxotrophs would have excreted CMs at different magnitudes, while the minimal quantity required of each CM to support cell growth would also be different.

Hence, to construct a successful complementary auxotrophic *S. cerevisiae* co-culture, taking the perspective of a *lys2ΔADE8* strain within a *lys2ΔADE8 - LYS2ade8Δ* co-culture[22] as an example, the cells need to i) have access to a sufficient pool of lysine during the initial phase of cultivation when lysine excretion by the complementary *LYS2ade8Δ* strain is minimal to ensure its cellular metabolism is not stalled (Fig. 3), which may be fulfilled through supplying the co-culture with small amounts of both lysine and adenine at the start of the process; and ii) carry sufficient *ADE8* gene dosage to ensure it is capable of excreting adenine at a quantity that will support the growth of the complementary *LYS2ade8Δ* strain beyond the initial phase after the small amounts of exogenously supplied lysine and adenine are depleted.

## Heterogeneity-reducing strategies for growth in non-selective media

Earlier, we showed that either reducing the capability of cells to import extracellular CMs or decreasing intracellular CMs biosynthesis by plasmid-carrying prototrophs would lead to improvement in plasmid persistence within a recombinant yeast population. Therefrom, we question if it is possible, through genetic engineering, to create recombinant yeast strains transformed with high-copy 2μ-plasmids that can attain a high plasmid retention rate even when cultivated in more nutritious, non-selective complex media, thereby taking full advantage of 2μ-plasmid's higher recombinant gene dosage to produce superior single-cell and ensemble productivity, in comparison to yeast with chromosomal-integrated genes that do not require selective media. This idea was exemplified in several antibiotic-free plasmid-

addiction systems that are independent of auxotrophic metabolite selection[45,54,69–73], though each of them has its own drawback(s), such as difficult to set up, inducing heavy growth penalty on host cells or was specifically tailored to the end-product-of-interest and thereby not broadly applicable.

Our approach was to investigate the combinatorial effect of deletion of high-affinity pyrimidine permeases in host chassis and employment of URA3-derived defective selection markers for plasmid retention to repress the growth of plasmid-free cells in non-selective YPD media. The late exponential-early stationary phase mTur2 expression profiles of recombinant yeast strains record reducing selective marker complementation efficiency in one dimension (URA3 > URA3d > URA3dCL1) and reducing pyrimidine uptake capability in the other dimension (BY4741> BY4741*fur4Δ*> BY4741*fur4Δ-fui1Δ*) are shown in Fig. 5 (Detailed temporal development of each strain's flow cytometry profile and their respective growth curve can be referred to Fig. S8). All modified strains showed superior plasmid-retention rate in YPD compared to the BY_mTU3 reference strain, with BYdbKO_mTU3d and BYdbKO_mTU3dCL1 maintaining 92.5% and 97.7% plasmid-carrying cells, respectively, after 46 h of batch cultivation. Nevertheless, the growth fitness of BY*fur4Δ*_mTU3dCL1 and strains with BY4741*fur4Δfui1Δ* background were hampered, observed as a much longer lag phase (Fig. S8B). As such, BY_mTU3dCL1 and BY*fur4Δ*_mTU3d, with plasmid retention rates of 84.9% (MFI = $2.5 \times 10^5$ a.u., rCV = 97.1) and 91.4% (MFI = $7.6 \times 10^4$ a.u., rCV = 118) (Fig. S8C), respectively, after 46 h of cultivation while having growth fitness comparable to that of the BY_mTU3 reference strain, would be the leading strains in terms of population ensemble recombinant protein productivity. Of note, in non-selective media, since cells no longer rely on the *URA3*-plasmids for survival, the deletion of pyrimidine permeases yields greater improvement in plasmid retention, as it reduces the relative growth fitness advantage of plasmid-free cells over plasmid-carrying cells.

## Co-cultivation systems built on the cross-feeding of complementary metabolites

Pivoting on the extensive CM cross-feeding in recombinant yeast populations carrying auxotrophic selection markers on high-copy 2μ plasmids, we hereby propose two conceptual mixed-culture designs that will exploit such phenomenon to establish and maintain synthetic mixed-cultures of yeast-yeast and yeast-bacteria that could be useful in instances wherein multi-strains and multi-species microbial cultures are preferred, for example when metabolic division of labour is favourable[74], where either design can be further optimised and tailored.

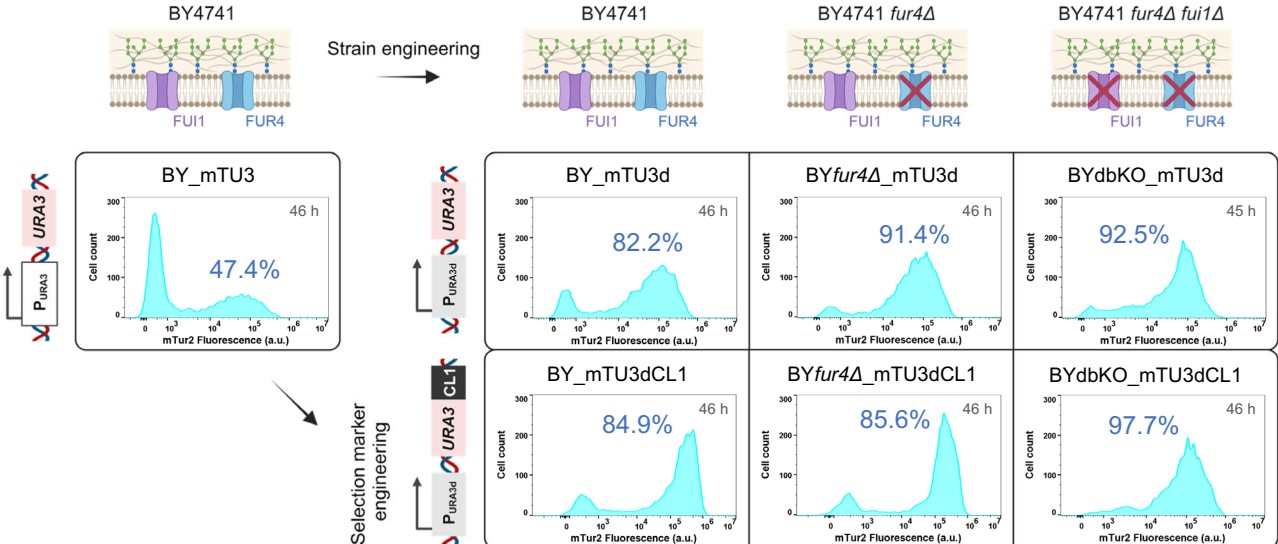

**Fig. 5 | The mTur2 expression profiles of various recombinant yeast strains at 45–46 h post-induction during batch mode fermentation in non-selective (YPD) media.** With the exception of BY_mTU3, the auxotrophic strains are complemented with plasmids harbouring either URA3d or URA3dCL1 selection marker. All yeast strains were seeded in corresponding selective media (SC-Ura for strains with BY4741 and BY4741*fur4Δ* cell chassis; SC-Ura-His for strains with BY4741*fur4Δfui1Δ* cell chassis) supplemented with 2% raffinose + 0.1% glucose prior to being induced in YPD media containing 2% galactose, wherein after 45–46 h the percentage of mTur2-expressing (%FP+ve) cells of each strain was reported. A single comparative experiment was conducted. Cartoon graphics were created with BioRender.com.

In the first scenario, we established a mixed culture consisting of a plasmid-complemented prototrophic *S. cerevisiae* strain BY*fur4Δ*_mTU3 and a pyrimidine auxotrophic *E. coli* strain BW25113*pyrFΔ*. A separate co-culture inoculated with BY*fur4Δ*_mTU3 and a prototrophic *E. coli* strain (BW25113) was used as a comparator. As shown in Fig. 6A, prototrophic *E. coli* cells dominated the yeast-bacteria co-culture throughout the 48 h fermentation. On the other hand, auxotrophic *pyrFΔ E. coli* did not outcompete co-residing yeasts but rather was maintained in proportion. In this design, the growth of auxotrophic BW25113*pyrFΔ E. coli* and plasmid-free derivative BY4741*fur4Δ* yeasts are constrained by the uracil and uridine being excreted via overflow metabolism from plasmid-carrying BY*fur4Δ*_mTU3 prototrophs, thereby forcing a stable, frequency-dependent co-culture between the three subpopulations. Such self-regulation capacity of a community may be useful for efficient division of labour designs. Cryptic growth of auxotrophic *E. coli* on lysed yeast cells was ruled out as <4% of yeast cells had compromised membranes at the 48-h sampling point (as measured with SYTOX Green, data not shown). This successfully established yeast-bacteria co-culture could be further improved by tuning the strength of the URA3 selection marker harboured by the BY*fur4Δ*_mTU3 plasmid-carrying yeast cells, as well as further optimisation of cultivation conditions.

In the second system design, a single strain-derived yeast mixed-culture was established by exploiting plasmid segregational loss and CM cross-feeding, with the aid of the commonly used Tet-R/Tet-O regulation system (Fig. 6B). In essence, BY4741 yeast was transformed with a high retention rate pKH−ve plasmid that harbours mCherry (mCh) and a defective LEU2d selection marker, along with a low retention rate pKH+ve plasmid to express mTurquoise2 (mTur2) under the maintenance of URA3 selection marker, to create the BY_KH strain. Driven by plasmid segregational loss and complementary metabolite cross-feeding, the original single-strain yeast population would morph into a 4-member mixed culture of mTur2+mCh+, mTur2−mCh+, mTur2+mCh− and mTur2−mCh−. The system has two induction switches embedded, with mTur2 under the tight regulation of pGal1, while cells carrying the pKH−ve would express a basal level of mCh under the regulation of pGal1[tetOx3] that is amplified by the addition of tetracycline (pGal1[tetOx3] was modified from P[tetO]3-in-GAL1 created

by Hsu and colleagues[75]). Under our design, mTur2 recombinant expression at the population level was initiated at the beginning of the batch fermentation and waned at a later growth phase, while mCh production was induced at a later growth state and maintained a more consistent recombinant expression output. Collectively, the current design allows both the division of labour among different sub-populations derived from the same yeast strain and a layer of temporal control in the induction of two different recombinant proteins. This design can be useful in applications that involve tandem reactions taking place within a single host chassis, where the required dosage of participant enzymes varies at different stages of batch fermentation. For instance, in biofuel biosynthesis using single-cell architecture[74], biomass feedstocks are firstly extracellularly digested into simplified sugars by enzymes, which are either secreted or displayed on the cell surface[76], prior to being consumed and intracellularly converted into biofuels in subsequent reaction steps (e.g., [77,78]). In such a scenario, it would be desirable to have the enzymes involved in sugar conversion be expressed in abundance at the beginning to quickly generate simple sugars, while the enzymes involved in converting sugars to biofuels are not required to have expression upregulated until a later stage of the bioprocess, i.e. when the simple sugars have accumulated and are no longer the limiting reagents.

### The significance of complementary metabolite cross-feeding beyond bioengineering

From our study, it is obvious that to effectively break a prototroph-auxotroph complementary metabolite (CM) cross-feeding linkage within a mixed population, it is crucial to address the matter from both aspects: reducing the endogenous overproduction of said CM in prototrophs to mitigate its overflow excretion, as well as minimise the capability of co-residing auxotrophs to scavenge said CM to support their cellular metabolism. Apart from bioprocessing and bioengineering in general, such findings also have important relevance to other fields of biological science, for instance, in the treatment of antimicrobial drug-resistant pathogens. In structured environments, such as cell colonies or biofilms, cross-feeding is enhanced by the limited diffusion rate of essential metabolites, thereby promoting heterogeneity within isogenic microbial populations through

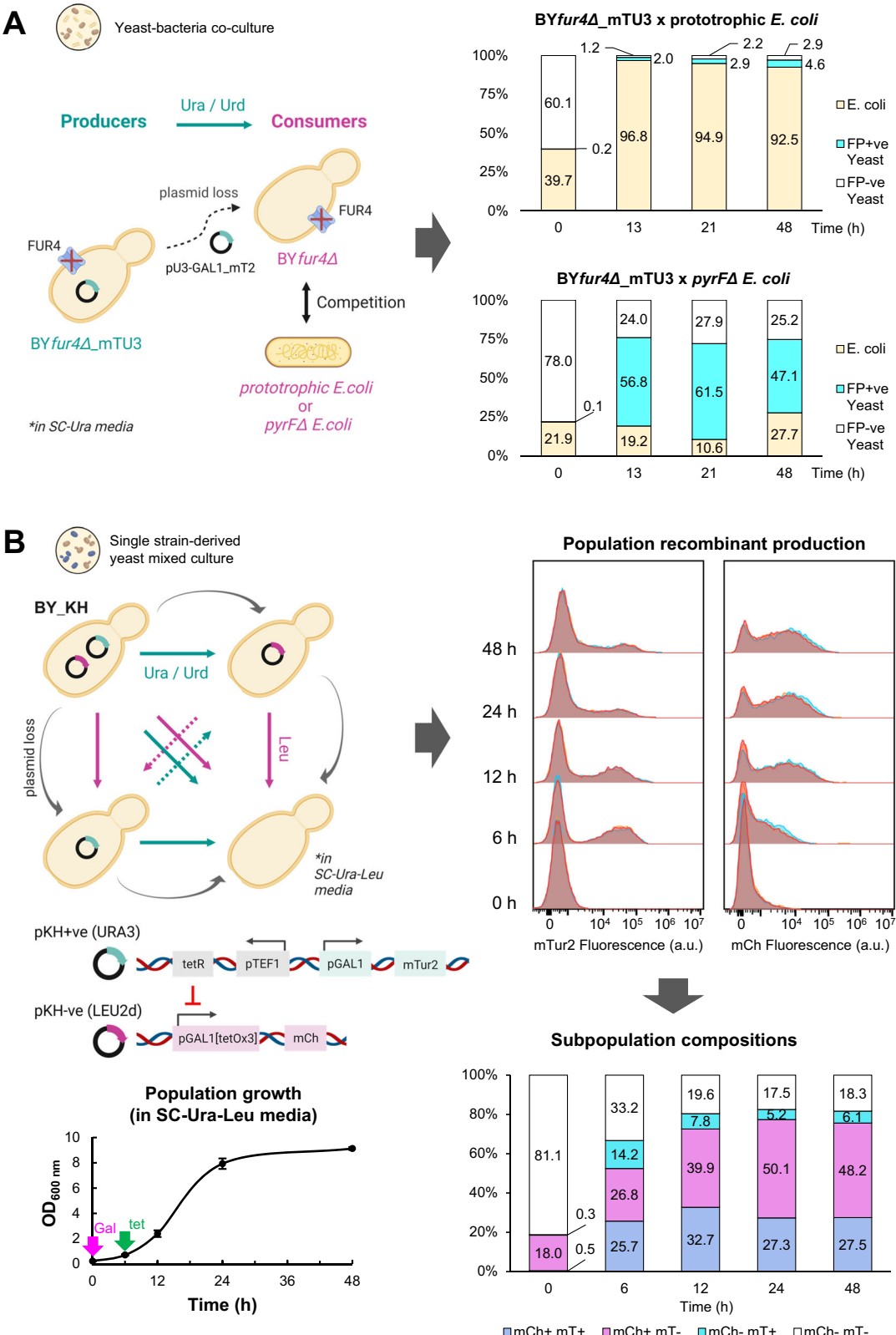

facilitating metabolic specialisation and eventually phenotypic diversification, leading to variations in antibiotic tolerance among isogenic cells[47]. Furthermore, essential metabolite exchanges also promote the survival and proliferation of auxotrophic mutants generated within the community, which can be more drug-tolerant in comparison to their prototrophic ancestor due to their higher metabolite efflux[9]. Cross-feeding can also modulate the antibiotic tolerance of a mixed-species

microbial community. For instance, pathogenic *Pseudomonas aeruginosa* relies on co-inhabiting with anaerobic bacteria to breakdown mucin while cross-feeding them with metabolites, and as a result, its tolerance to ampicillin was lower than when it was grown in mono-culture since the mucin-degrading bacteria were more sensitive to the drug[79]. In another example of co-infecting pathogens, *Acinetobacter baumannii* was shown to cross-protect *Klebsiella pneumonia* against

**Fig. 6 | Synthetic microbial co-culture designs exploiting complementary metabolite(s) cross-feeding under auxotrophic selection pressure. A** Left: Yeast −*E. coli* co-culture design wherein plasmid-free, pyrimidine auxotrophic yeasts and *E. coli* (either prototrophs or pyrimidine auxotrophs) compete for pyrimidines, namely uracil (Ura) and uridine (Urd), excreted by plasmid-carrying prototrophic yeasts into a modified SC-Ura media (containing 2% galactose and 1% raffinose) over 48 h of batch mode cultivation. Right: Cell culture compositions, as determined via flow cytometry, when BY*fur4Δ*_mTU3 yeasts were mix-cultivated with either prototrophic or pyrimidine auxotrophic (*pyrFΔ*) *E. coli*. *n* = 2, bio-replicates are presented in Fig. S9. **B** Design of a single strain-derived yeast mixed culture system wherein the strains have built-in division-of-labour features upon plasmid loss, hedging on the Ura/Urd cross-feeding potentials provided by URA3, and to a much lesser extent leucine (Leu) by LEU2d, selection markers, as well as the tetR−tetO regulatory system (top left). Population growth curves are presented at the bottom left (growth data points were presented as mean values ± s.d.). Population recombinant expression profiles (top right) and compositions (bottom right) upon induction of the cell cultures (galactose at *t* = 0 h for mTur2 and tetracycline at *t* = 6 h for mCherry) were determined via flow cytometry. *n* = 3, bio-replicates are presented in Fig. S10; (**A**) and (**B**) Note that at *t* = 0 h no fluorescent protein was present in yeasts carrying mTU3 plasmids since they were un-induced. Cartoon graphics were created with BioRender.com.

cephalosporin, while the latter cross-feed it with by-products of sugar fermentation in return[80]. Henceforth, for effective antimicrobial treatment, it could be important to track the cross-feeding relationships between constituent species within a drug-resistant mixed population of microbials and identify the "weakest" link among them, which can then be targeted and sabotaged.

Another research area wherein intercellular cross-feeding and its targeted mitigation can have contributory impacts is cancer treatment. Specifically, amino acid depletion strategies are emerging as promising alternatives to conventional chemotherapies in order to improve the quality of life of cancer patients who receive long-term treatment. Now, asparaginase has become part of the routine paediatric acute lymphoblastic leukaemia multi-agent therapy regime, and many other therapies involving either asparaginase, arginase, or glutaminase are under clinical trials, all aimed at reducing the availability of exometabolome amino acids that are essential for certain cancer cells, thereby starving cancer cells to death [reviewed in[13]]. Nevertheless, it was shown that surrounding healthy cells can cross-feed cancer cells with these essential metabolites[81,82]. In addition, bacteria and fungi are also known to take up residence in solid tumours both extracellularly and intracellularly[83,84], providing another avenue for cancer cells to exchange the essential metabolites required for growth. Both of the aforementioned points make it particularly challenging for amino acid depletion treatment against solid tumours. Drawing from our study, we postulate that it is necessary to identify and address potential cross-feeding linkages of essential metabolite(s) between cancerous cells and co-existing healthy cells and microbes in order to improve the outcome of amino acid depletion therapies, which could be achieved through genetic engineering methods, for instance via the co-administration of microRNAs (miRNAs) or small interfering RNAs (siRNAs) therapies[85].

## Methods

### Plasmid and strain construction
Plasmids used in this study were synthesised, and codon optimised for recombinant expression in *S. cerevisiae* or *E. coli*, as appropriate, by GenScript Biotech (Singapore) and are listed in Table S1. The yeast and *E. coli* strains constructed are listed in Table S2. All yeast plasmids were built on vectors from the pESC series, with modifications on promoter sequence and selection marker being outlined in Table S3.

The *S. cerevisiae* BY4741 parental strain was gifted by Prof. Hongyuan Yang of the University of New South Wales (Australia). The yeast strains yAG1, yAG2, yAG19, and yAG20 were gifted by Assistant Prof. Andrea Giometto of Cornell University (USA) and Prof. Andrew Murray of Harvard University (USA)[86]. The yeast strains BY4741*urk1Δ* and BY4741*fur4Δ* from the Yeast Knockout (YKO) Collection[48,87], as well as the *E. coli* parental strain BW25113 and the BW25113*pyrFΔ* strain from the *E. coli* Keio Knockout Collection[88], were all sourced from Horizon Discovery (UK). The BY4741*fur4Δfui1Δ* strain was constructed by replacing the *FUI1* gene of the BY4741*fur4Δ* strain with the *HIS3* selection marker through homologous recombination. Positive clones were then selected on SC-Ura-His agar plate supplemented with 4 mM cytosine (Sigma-Aldrich, Australia) and 2.5 mM 5-fluorouridine (Tokyo

Chemical Industry, Japan). Primers being used to create the DNA fragment to knock out the *FUI1* gene and later on for the screening (through PCR) and verification (through Sanger sequencing) of positive clones are listed in Table S4.

Yeast strains were transformed following the DTT/ LiAc-SS carrier DNA/ DMSO/ PEG transformation procedures[89,90] using the Yeast Transformation Kit (Sigma-Aldrich), and transformants were selected on appropriate SC selective agar plates without the required supplement (either SC-Ura, SC-Leu, SC-Ura-Leu, or SC-Ura-His). Of note, all BY4741*fur4Δfui1Δ* strains were selected on SC-Ura-His agar containing 400 μg/L G418 (Sigma-Aldrich) and 2.5 mM fluorouridine. In addition, all strains carrying the pU3dCL1-GAL1_mT2 plasmids were first inoculated into SC-Ura liquid medium containing 2% (w/v) glucose immediately after the transformation procedures and left at room temperature for 2-4 weeks (until the cell culture turned slightly turbid) to allow the transformants to adapt to the auxotrophic selection pressure prior to plating on either SC-Ura or SC-Ura-His selective plates, as appropriate, for single colony isolation.

### Yeast cultivation
Yeast cells were cultivated in either complex YPD medium or synthetic drop-out SC-X medium (wherein X is either Uracil, Leucine or Histidine drop-out, or a combination of them). YPD was prepared with 10 g/L of Yeast extract (Millipore), 20 g/L of Bacto™ Peptone (BD Biosciences) and 20 g/L of either glucose or galactose (Sigma-Aldrich). SC-X synthetic drop-out media were prepared with 6.7 g/L of Yeast nitrogen base without amino acids (Sigma-Aldrich) along with the required amount of appropriate Yeast synthetic drop-out medium supplements (from either Sigma-Aldrich or Formedium, 1.92 g/L for -Ura; 1.62 g/L for -Leu, 1.54 g/L for -Leu-Ura, 1.85 g/L for -Ura-His) and carbon sources. SC synthetic complete media was prepared by adding 76 mg/L of uracil (Sigma-Aldrich) into the SC-Ura formulation. When antibiotics were added into a synthetic drop-out medium, 6.7 g/L of YNB without amino acids was replaced with 1.7 g/L of YNB without amino acids and ammonium sulphate (Formedium) plus 1 g/L of monosodium glutamate (Sigma-Aldrich). Solid agar was prepared by adding 20 g/L bacteriological grade agar powder (Thermo Fisher) to the liquid formulation.

Unless noted otherwise, auxotrophic yeast strains were initially seeded in SC-X seed media supplemented with 2% (w/v) raffinose (Sigma-Aldrich) plus 0.1% (w/v) glucose and incubated at 30 °C and 220 rpm in INFORS HT Multitron incubator (Infors AG) and grown to exponential phase. Subsequently, the exponentially-growing cells were added into induction media, which was either SC-X supplemented with 2% (w/v) galactose plus 1% (w/v) raffinose or YPD with 2% (w/v) galactose to an initial culture $O.D._{600\ nm}$ of 0.2−0.4 (measured using CO8000 Cell Density Metre, Biochrom) to induce the expression of recombinant genes under the regulation of $P_{GAL1}$ and $P_{GAL10}$. The recombinant protein expression stage was typically performed in 10−20 mL media in 100 mL Erlenmeyer flasks.

In the experiment evaluating the phenotypic diversification of the BY_KH strain due to progressive plasmid segregational loss in SC-Ura-Leu induction media, 100 μM of tetracycline was added into the cell

cultures 6 h after galactose induction to alleviate the repression of tetR protein against the pGal1[tetOx3] promoter.

## Yeast and *E. coli* co-cultivation

The *S. cerevisiae*–*E. coli* co-cultivation experiment was conducted in modified SC-Ura medium (SC-Ura medium plus 20 g/L galactose, 10 g/L raffinose, 15 g/L KH$_2$PO$_4$, 2.5 g/L NaCl, 33.9 g/L Na$_2$HPO$_4$ and 5 g/L NH$_4$Cl), wherein phosphate salts were added to increase the buffering capacity of the medium against the acidification of secreted metabolites (in particular ethanol[91]) that will impede the growth of *E. coli* within the co-culture. The *E. coli* seed cultures were prepared in Terrific Broth (TB) medium and grown at 37 °C, 200 rpm overnight. The BY*fur4Δ*_mTU3 *S. cerevisiae* seed culture was prepared in SC-Ura medium and grown at 30 °C, 220 rpm overnight. Subsequently, 400 μL of yeast-grown cell suspension of O.D.600 $_{nm}$ ~ 6.4 was inoculated into 5 mL of modified SC-Ura medium in 50 mL Falcon tube, and 10 μL of *E. coli* seed (O.D.600 $_{nm}$ ~ 4.7 and 6.5 for BW25113 and BW25113*pyrFΔ*, respectively) was added into the yeast culture. The yeast–*E. coli* co-cultures were then incubated at 30 °C, 200 rpm for 48 h.

## Flow cytometry

Flow cytometry analysis was performed using either the CytoFLEX V5-B3-R3 or CytoFLEX LX V5-B3-Y5-R3 flow cytometer [Beckman Coulter, lasers: 405 nm (V), 488 nm (B), 561 nm (Y) 638 nm (R)]. Quality control checks and blanking with phosphate-buffered saline (PBS) were performed to subtract any background noise from the measured samples. The acquisition threshold was set to be based on forward scattering value (FSC) under automatic mode. Cell samples were analysed at a flow rate of 10 μL/min until at least 20,000 events were recorded. Flow cytometry data was acquired with CytExpert software v2.3 and v2.5 (Beckman Coulter) and processed with FlowJo software v10.6.1–v10.8.0 (FlowJo LLC). Samples were cell count-normalised to 10,000 events (unless otherwise stated) using the DownSampleV3 plugin within FlowJo. The subpopulation compositions of yeast monocultures were assessed using the gating strategy illustrated in Fig. S11, while the compositions of yeast–*E. coli* co-cultures were analysed using the gating strategy illustrated in Fig. S12.

Cell size was estimated through the forward scattering (FSC-A) value. FPs were detected using the following combination of laser and detector(s): mTurquoise2−405 nm and PB450-A (450/45 nm); mVenus and mCitrine−488 nm and FITC-A (525/40 nm); mKOκ−488 nm and PE-A (585/42 nm); mCherry−either 488 nm and ECD-A (610/20 nm), 488 nm and PC5.5-A (690/50 nm), or 561 nm and mCHERRY-A (610/20 nm).

Cell samples were pre-stained with either propidium iodide (PI, 5 μg/mL final concentration, Thermo Fisher) or SYTOX™ Green Nucleic Acid Stain (SYTOX, 1 μM final concentration, Thermo Fisher) depending on the FPs being expressed (PI for mTurquoise2; SYTOX for mVenus, mCitrine, mKOκ, and mCherry). Only FP signals from healthy, viable cells (as inferred through their intact cell membrane and hence PI-negative and SYTOX-negative status) were reported. The PI signal was monitored using a 488 nm laser and PC5.5-A channel, while the SYTOX signal was monitored using a 488 nm laser and either a FITC-A channel (for mKOκ and mCherry) or PC5.5-A channel (for mVenus). In addition, cell debris, aggregates and cell doublets were also gated out from the acquired flow cytometry data[92].

The mitochondrial membrane potential (Δψm) of yeast cells was assessed using the TMRE-Mitochondrial Membrane Potential Assay Kit (Abcam). Specifically, TMRE (tetra-methyl-rhodamine ethyl ester) in DMSO was added to unfixed cells suspended in PBS buffer (100 nM final concentration) incubated in the dark at 30 °C with shaking for 30 min prior to analysis using the PE-A channel. A Δψm depolarisation control sample was created by treating cells with FCCP (carbonyl cyanide 4-(trifluoromethoxy) phenylhydrazone) in DMSO (20 μM final concentration) for 10 min in dark at 30 °C with shaking prior to TMRE staining.

## Fluorescence-activated cell sorting

FACS was performed using a BD Influx System (equipped with 405, 488, 561 and 635 nm lasers, BD Biosciences) at Monash FlowCore Platform via the BD FACS™ Software version 1.2.0.142. The details of the cell sorting strategy performed are illustrated in Fig. S2A. Firstly, PI +ve cells were excluded. Subsequently, cells at the bottom 20% in mTur2 fluorescence intensity (FI, a.u.) of the FP−ve subpopulation were sorted into the "mTur2−ve" sample; cells at the top 20% in mTur2 FI (a.u.) of the FP+ve subpopulation were sorted into the "mTur2+ve" sample; a sample population that was PI−ve but ungated against mTur2 FI was sorted into the "FACS Ctrl" sample. Details of the instrument settings and sorted samples are included in Table S5.

## Study of prototrophic and pyrimidine auxotrophic yeast cell growth in SC-Ura spent media

BYmTU3 cells were grown in SC-Ura media containing 2% (w/v) galactose and 1% (w/v) raffinose for 48 h. Spent "SC-Ura" media was collected via centrifugation (7378*g*, 10 min) and filtered through 0.2 μm filters three times to ensure complete removal of BYmTU3 cells, which was verified through flow cytometry. Subsequently, the spent media was re-conditioned by adding the following ingredients to the specified final concentration: 13.15 g/L NaH$_2$PO$_4$, 1.254 g/L Na$_2$HPO$_4$, 20 g/L galactose, 10 g/L raffinose, 1 g/L monosodium glutamate. This was both to replenish the carbon and nitrogen sources within the spent media, as well as to bring it to a pH range that is suitable for yeast cell growth (the reconditioned media had a pH of ~4.5).

BY4741 cells pre-grown overnight in YPD media containing 2% galactose were harvested via centrifugation (7378*g*, 10 min), washed twice with PBS buffer prior to being inoculated into either freshly prepared SC-Ura media or the re-conditioned "SC-Ura" spent media to an initial culture O.D.$_{600\ nm}$ of ~0.2 and 0.3, respectively. The cultures were subsequently incubated at 30 °C, 220 rpm for 48 h. The use of galactose as a carbon source in the reconditioned "SC-Ura" spent media allowed the verification using flow cytometry of the absence of any remnant BYmTU3 FP+ve, pyrimidine-producing cells in the spent media that could otherwise confound the observed growth of the BY4741 cell cultures.

Concentrations of pyrimidines within the sampled yeast culture spent media were measured using the protocol developed by Hou and Ding[93]. Spent media (0.4 mL) was collected from yeast culture samples after being centrifuged at 16000*g* for 2 min and syringe-filtered into glass HPLC vials using a 0.45 μm nylon filter. Duplicate samples (3 μL) were injected into an Agilent 1220 HPLC system (Agilent Technologies) equipped with a Luna C18 column (250 mm × 4.5 mm, 5 μm) and guard column supplied by Phenomenex, NSW, Australia and ChemStation LC data software (Agilent Technologies). Calibration curves established for each pyrimidine were presented in Table S6; example chromatograms of a pyrimidine standard mixture and a yeast culture spent media sample were included in Figure S13.

## Population productivity ensemble measurement using TECAN

Population-average fluorescence intensity of yeast cell culture was measured using Infinite® 200 PRO microplate reader (Tecan, Switzerland). Cell samples were diluted to an O.D.$_{600\ nm}$ of 0.4 and 0.2 mL of which were dispensed into each well. Absorbance was measured at 600 nm, and fluorescence was measured at 475 nm. For each measurement, each well was read 9 times and an average value was taken.

## Reporting summary

Further information on research design is available in the Nature Portfolio Reporting Summary linked to this article.

## Data availability
Source data are provided as a Source Data file. Raw and processed flow cytometry data are available upon request to the authors. Source data are provided with this paper.

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

## Acknowledgements

K.H. received RTP and GRCA scholarships from Monash University. This work was also supported by the ARC LIEF grant No. LE16100185. We express gratitude to Traude Beilharz, Michael McDonald and Andrea Giometto for their valuable discussions and advice, as well as technical support from Brian Jong, Xueqing Liu, Zongsheng Zou and Tayyaba Younas. We thank Monash FlowCore for assistance with FACS and Monash Micromon for gene sequencing services. Cartoon graphics used in the figures were created with BioRender.com.

## Author contributions

K.H., G.D. and V.H. conceived the study. G.D. and V.H. supervised the study. K.H. and V.H. designed the experiments. K.H. carried out the experiments with the assistance of A.S., and K.H. analysed data and prepared figures. K.H. and V.H. wrote, revised and edited the paper. All authors approved the final paper.

## Competing interests

The authors declare no competing interests.
