## [Peer Review File · Nature Communications]

Cross-feeding promotes heterogeneity within yeast cell populationsEditorial Note: This manuscript has been previously reviewed at another journal that is not operating a transparent peer review scheme. This document only contains reviewer comments and rebuttal letters for versions considered at *Nature Communications*.

Reviewer #1 (Remarks to the Author):

The authors have adequately addressed my concerns in the first review.

Reviewer #2 (Remarks to the Author):

I appreciate the authors' efforts in addressing previous concerns. However, the novelty and impact of this manuscript are still limited. In their response letter, the authors agreed that metabolite cross-feeding in bacteria-to-bacteria and yeast-to-bacteria systems are well studied. The revised manuscript argues that the novelty lies in "syntrophic growth between complementary yeast auxotrophs". Yet, there are several prior studies on yeast-to-yeast cross-feeding of essential metabolites (e.g. [10.1073/pnas.0610575104](https://doi.org/10.1073/pnas.0610575104), [10.1073/pnas.1313285111](https://doi.org/10.1073/pnas.1313285111), [10.1371/journal.pbio.1002540](https://doi.org/10.1371/journal.pbio.1002540), in addition to the references provided by the authors). While this manuscript demonstrates that cross-feeding promotes plasmid-free cells under an auxotrophic selection system, this issue specifically arises with the URA3-containing plasmid. This specific problem can be easily avoided by chromosomal integration of the target gene or using alternative selection systems without cross-feeding, especially if metabolite cross-feeding is not widespread in yeast as the author stated.

Overall, the manuscript lacks conceptual novelty given the extensive prior observations of plasmid loss caused by cross-feeding in bacterial systems. The addressed issue seems to be specific, unless the authors can demonstrate that cross-feeding is a widespread phenomenon across various commonly used yeast plasmid systems with different selection mechanisms. The current manuscript is better suited for a more specialized journal.